# Effects of Iron Oxide Nanoparticle Supplementation on the Growth Performance, Serum Metabolites, Meat Quality, and Jejunal Basal Morphology in Broilers

**DOI:** 10.3390/ani14010099

**Published:** 2023-12-27

**Authors:** Sara Basharat, Sajid Khan Tahir, Khalid Abdul Majeed, Muhammad Shahbaz Yousaf, Khalil Khadim Hussain, Muhammad A. Rashid, Hafsa Zaneb, Habib Rehman

**Affiliations:** 1Department of Physiology, University of Veterinary and Animal Sciences, Lahore 54000, Pakistan; sarabasharat89@gmail.com (S.B.); drmshahbaz@uvas.edu.pk (M.S.Y.); habibrehman@uvas.edu.pk (H.R.); 2School of Applied Sciences, University of Brighton, Brighton BN2 4GJ, UK; k.hussain@brighton.ac.uk; 3Department of Animal Nutrition, University of Veterinary and Animal Sciences, Lahore 54000, Pakistan; drafzal@uvas.edu.pk; 4Department of Anatomy and Histology, University of Veterinary and Animal Sciences, Lahore 54000, Pakistan; hafsa.zaneb@uvas.edu.pk

**Keywords:** production performance, trace minerals, health biomarkers, jejunal histology, poultry

## Abstract

**Simple Summary:**

Iron is a necessary micronutrient in animal feed. Iron is an important component of hemoglobin, myoglobin, and the enzymes participating in redox reactions. The aim of the current study was to assess the effects of iron oxide nanoparticle supplementation on production performance, organ development, blood biochemistry, redox status, meat quality, and jejunal histology in broilers. The results indicated that iron oxide nanoparticle supplementation improved the feed conversion ratio, pectoral muscle, and jejunal histology. However, iron oxide nanoparticle supplementation showed no effects on visceral organ development, blood metabolites, redox status, and carcass traits.

**Abstract:**

The current research aimed to evaluate the supplemental effects of iron oxide nanoparticles (IONPs) on production performance, viscera development, blood metabolites, redox status, meat quality, and jejunal histology in broilers. A total of 300 day-old broilers were randomly divided into six groups with five replicates per group. Birds were fed on a corn soybean-based diet supplemented with 0, 20, 40, 60, or 80 mg/kg IONPs or 80 mg/kg of FeSO_4_ for 35 days. The feed conversion ratio (FCR) was improved in birds supplemented with 60 mg/kg IONPs. The pH_24h_ was lower in birds supplemented with 40 mg/kg IONPs compared to that of the bulk group. Pectoral muscle fascicle diameter and fiber density were significantly increased in 20 mg/kg IONP-supplemented birds compared to those of the bulk group, respectively. The muscle fiber diameter was higher in 40 mg/kg IONP-supplemented birds compared with the bulk group. The jejunal villus height, crypt depth, and villus surface area were significantly increased with 60 mg/kg IONP supplementation, whereas villus width was decreased in birds supplemented with 40 mg/kg IONPs. The villus-height-to-crypt-depth ratio was lower in IONP-supplemented birds compared to the bulk group. IONP supplementation improved the FCR, jejunal, and pectoral muscle morphology without affecting the carcass characteristics and redox status of broilers.

## 1. Introduction

Iron is an important trace element that plays a vital role in many metabolic processes in human and animal organisms. It is added to compound feeds and premixes in the form of inorganic salts, organic forms, or chelates. Iron is an important component of hemoglobin, myoglobin, and the enzymes involved in redox processes [1]. The absorbability of iron in various forms is decreased by its insoluble or poorly soluble nature. Therefore, it is essential to complement the diet with iron [2]. According to the most recent study, broiler breeder hens need an average of 106 ppm of iron [3]. Replacing FeSO_4_ with Fe-Methionine in broiler diets was shown to increase the breast meat yield [4]. Animals require higher amounts of mineral supplements since they not only use minerals at the cellular level but also excrete a large number of them into the environment, which raises the feed’s cost. Feeding traditional iron preparations is also justified by the low gut absorption of iron [5].

With the advancement of nanotechnology, the use of manufactured nanoparticles (100 nm or less) as dietary supplements expanded. Nano-feed additives not only improved feed efficiency but also lowered feed costs with the added benefit of enhancing animal production [6]. Numerous studies reported on the use of various nanoparticles including copper [7], zinc [7], selenium [7], and iron [3], where the total dose recommended to achieve favorable serum concentrations at nano levels was less than the required levels for their normal substitutes [8]. According to research by Jain et al. [9], long-term administration of magnetic iron oxide nanoparticles is both hepato-protective and safe in terms of oxidative stress. Commercially available iron oxide nanoparticles have been employed in numerous toxicity and immunological studies [10].

The beneficial effects of iron nanoparticles on productivity and blood parameters in broilers were studied [11]. Nanoparticles enhanced the health status of livestock and poultry by improving feed intake, nitrogen retention, oxidative status, microbial biomass, and by reducing the effects of heat stress. Iron oxide nanoparticles have also been identified in the tissues of mice as showing peroxidase-mimicking activity, which is based on its capacity to catalyze oxidation processes [10]. The use of iron nanoparticles in broilers to assess the growth performance, serum biochemical, and meat quality attributes at comparable levels has only been the subject of a small number of investigations. Therefore, the current study was intended to assess the effects of the supplementation of IONPs on growth performance, viscera development, meat quality, redox status, serum metabolites, and jejunal histology in broilers.

## 2. Materials and Methods

### 2.1. Preparation and Characterization of Iron Oxide Nanoparticles

Briefly, iron oxide nanoparticles (IONPs) were prepared in an aqueous medium by the addition of 0.25 M iron chloride (Daejung, Busan, Republic of Korea) to 0.25 M iron sulphate (Daejung, Republic of Korea). The mixture was allowed to stir on a hot plate magnetic stirrer (DAIHAN Scientific Co., Ltd., WISD Laboratory Instruments, Wonju, Republic of Korea) for 30 min under a nitrogen atmosphere at 45 °C to remove any dissolved oxygen [12]. Thereafter, 6 M Sodium Hydroxide (Sigma Aldrich, St. Louis, MO, USA) was added to the above mixture and stirred constantly for a further 40 min until a brown precipitate was formed. Finally, the precipitate was centrifuged for 8 min at 6000 rpm, (HARRIER 18/80 REFRIGERATED centrifuge, Handerson Biomedical, Ltd., Croydon, UK), washed with deionized water four times to remove any excess ammonia, and then directly dried in an oven (WISEVEN Fuzzy control system Wis Laboratory Instruments) at 60 °C for 24 h. Furthermore, the IONPs were characterized using an X-ray diffractometer (XRD) and transmission electron microscopy (TEM). To identify the crystalline phases and chemical composition information of the prepared samples, XRD analysis was performed under the following conditions. The diffractograms were recorded at room temperature with Cu as the tube anode, the steps were 0.02° of (°2θ), the counting rate was 0.5 s/step, the current was 30 mA, and the voltage was 45 kV. Data were collected using a scattering angle (2θ) ranging from 4° to 80°, as shown in Figure 1a. The XRD patterns of the prepared iron oxide nanoparticles showed six characteristic peaks of iron oxide at 2θ values of 30.1, 35.5, 43.3, 53.7, 57, and 62.8, as shown in Figure 1a. Similar results were previously reported by Ghasemi et al. [13]. The morphological analysis and size of the iron nanoparticles were estimated by transmission electron microscopy. Prior analysis of the samples was prepared as follows. A droplet from the sample was placed on a carbon-coated copper grid, and the excess fluid was removed by an absorbent filter paper and allowed to dry. The obtained TEM image from the IONPs showed spherical-shaped nanoparticles. TEM images of the uncoated iron oxide nanoparticles in Figure 1b demonstrate spherically shaped particles with an average diameter of 9.42 ± 1.12 nm, which is in good agreement with the previous report [14].

### 2.2. Experimental Design and Husbandry

A total of 300, one-day-old broiler (Hubbard) chicks, purchased from a local hatchery, were randomly divided into six groups (50 birds/group), each group having 5 replicates (10 birds/replicate). Birds in the first group were labeled as control birds and offered a basal diet without iron. Similarly, birds in the second group were labeled as bulk iron and were offered a basal diet with the addition of a bulk form of ferrous sulfate @ 80 mg/kg of feed. Birds in the remaining four groups were labeled as 20-IONPs, 40-IONPs, 60-IONPs, and 80-IONPs and offered basal diets with the addition of graded levels of IONPs at concentrations of 20, 40, 60, and 80 mg/kg of feed, respectively, for 35 days. The feed and water were provided ad libitum to the birds. Temperature and relative humidity on day 1 were kept at 35 ± 1.1 °C and 65 ± 5%, respectively. The temperature was decreased by 3 °C each week until it reached 26 °C. The composition of the diet is presented in Table 1.

### 2.3. Production Performance

Production performance was assessed by measuring the daily feed intake, weekly bodyweight gain, and feed conversion ratio. Feed intake was measured by getting the difference between feed offered and feed consumed. Bodyweight gain was measured by subtracting the initial bodyweight from the final bodyweight, while the feed conversion ratio was calculated by dividing the feed intake by the bodyweight gain.

### 2.4. Samples Collection and Preservation

On the 35th day, after an overnight fasting, two birds from each replicate were randomly selected and slaughtered. Blood samples were drawn and permitted to clot to extract the serum, then kept at −40 °C until further analyses. Samples of liver and pectoral muscle were removed, washed with normal saline, and stored at −40 °C for the redox status analyses. For histology, mid-jejunum and pectoral muscle were collected and preserved in 10% neutral buffered formalin.

### 2.5. Viscera Development

To determine the relative weights and lengths of the viscera, post-slaughtering, the viscera were removed and the weights of the liver, pancreas, gizzard, proventriculus, heart, small intestine, caecum, and immune organs (spleen, cecal tonsils, and bursa of fabricius) were determined with a digital weight balance, and the length of the small intestine and caecum was measured with a measuring tape.

### 2.6. Blood Metabolites

For biochemical analyses, serum samples were thawed, mixed, and then subjected to the determination of a lipid profile (total cholesterol, triglycerides, HDL-C, and LDL-C), a muscle function test (creatinine kinase and LDH), kidney function test (creatinine, urea, and uric acid), liver enzymes (ALT and AST), and serum proteins (total proteins, albumins, and globulins). These blood metabolites were determined by using commercially available kits (Diasys, Holzheim, Germany) according to the manufacturer’s instructions, with the help of an EPOCH^TM^ microplate spectrophotometer (Biotek Instruments Inc., Winooski, VT, USA).

### 2.7. Redox Status

Redox status in the serum, liver, and pectoral muscle was determined by the estimation of malondialdehyde (MDA) and catalase activity according to the protocols described by AMDCC Protocols [15] and [16], respectively.

### 2.8. Meat Quality

The carcass weight was determined after the removal of feathers and blood. The eviscerated weight was determined after the removal of the head, feet, and organs except the lungs and kidneys. A percentage of bodyweight was then used to express the carcass yield [17]. For dressing percentage, each bird’s breast muscle, thigh muscle, and abdominal adipose tissues were removed and weighed [18]. The pH_45min_ and pH_24h_ of pectoral muscle after 45 min and 24 h of slaughtering were determined by a pH meter (PCE-228 M pH meter, Southampton, UK) with a conventional probe electrode which was inserted into muscle tissue [19].

The water-holding capacity (WHC) of the pectoral muscle was determined by the method described by [20]. Briefly, samples were centrifuged at 500 g for 15 min at 4 °C. WHC (%) was computed by using the equation: 100 − [(Wi − Wf/Wi) × 100], where Wi and Wf are the initial and final sample weight, respectively. For cooking loss, the samples of pectoral muscle were weighed and placed in plastic bags. The bags were then placed in a pre-heated water bath set at 80 °C. When the bags’ internal temperature reached 78 °C, they were carefully blotted dry with paper towels without squeezing and allowed to cool by being submerged in running water for 30 min.

Then, the muscle samples were taken out of the polyethylene bags and weighed again to measure the cooking loss percentage [21]. Using a chromameter (CR-410, Konica Minolta, Tokyo, Japan), the color of pectoral muscle was measured 45 min after slaughter to obtain CIE values with lightness (L*), redness (a*), yellowness (b*), Hue (H), and chroma (C) [16]. For muscle histology attributes (muscle fascicle diameter, muscle fiber diameter, and muscle fiber density), pectoral muscles from two birds per replicate were collected, fixed in 10% neutral buffered formaldehyde, and processed according to a conventional method of hematoxylin and eosin followed by examination with a light microscope (Olympus CX31, Olympus USA, Center Valley, PA, USA) fitted with a digital-imaging system (Olympus DP20, Olympus USA). For morphological measurements, three complete fascicles were chosen from images obtained at 4×, and their vertical and horizontal measures were utilized to estimate the diameter of the muscular fascicles. The density of muscle fibers was measured in a 0.5 mm-radius circle created on the images captured at 4×. After dividing the circle in half, the number of fibers inside the right half were determined. The slide images captured at 10× were used to measure the diameter of the muscle fibers. On each image, a rectangle with the same dimensions was created, and it was then divided into two rows and five columns (10 boxes). Muscle fiber diameter was calculated by averaging the vertical and horizontal diameters of the fibers that were found in the alternating boxes [22].

### 2.9. Jejunal Histology

The jejunal micro-architecture was analyzed by measuring the villus height, villus width, crypt depth, villus surface area, and villus-height-to-crypt-depth ratio. Briefly, segments of jejunum were collected from birds (two birds per replicate), fixed in 10% neutral buffered formaldehyde, processed, and then stained according to the conventional staining method of hematoxylin and eosin. Finally, the slides were examined with a light microscope (Olympus CX31, Olympus USA) fitted with a digital-imaging system (Olympus DP20, Olympus USA) for histomorphometry. For measurements, two sections of jejunum per bird were taken and, in each section, five villi with well-oriented and intact lamina propria were selected. The measurement of villus height was taken from the tip to the villus crypt junction, while the measurement of crypt depth was taken from the base to the area where the crypt and villus transition. The formula (2p) (villus width/2) (villus length) was used to calculate the villus surface area [23].

### 2.10. Statistical Analysis

The data were checked for normality by using the Kolmogorv–Simov test. The replicate pen was considered as an experimental unit for all traits. Then, the data were subjected to a one-way analysis of variance using the Statistical Package for Social Sciences (SPSS for Windows version 22, IBM, Armonk, NY, USA). For group differences, Tukey’s test was used. Polynomial contrasts were used to determine the linear and quadratic effects of different levels of IONPs at *p* < 0.05, and the control group (0 IONPs) was included in the polynomial contrast analysis along with the various levels of IONPs used.

## 3. Results

### 3.1. Effects of Iron Oxide Nanoparticles Supplementation on the Growth Performance of Broilers

The bodyweight gain remained unaffected by bulk iron or IONP supplementation during the first three weeks of the experiment. However, during week 4, an increase (*p* < 0.05) in bodyweight gain was observed in the birds supplemented with 60 mg IONPs/kg compared to the control group. Also, significant linear and quadratic effects on bodyweight gain were observed in week 4 (Table 2). No effects on feed intake were found in all the birds throughout the experiment (Table 2). During the first week, FCR was higher (*p* < 0.05) in the birds supplemented with 80 mg/kg IONPs compared to the other supplemented and control groups. However, during the fifth week, improved FCR (*p* < 0.05) was observed in the birds supplemented with 60 mg/kg IONPs compared to the control, bulk iron, and 20, 40, and 80 mg/Kg IONP-supplemented groups. Also, the linear and quadratic effects on FCR were observed in week 1 and 5 (Table 2).

### 3.2. Effects of Iron Oxide Nanoparticles Supplementation on the Viscera Development of Broilers

The relative viscera weights (liver, heart, gizzard, proventriculus, spleen, bursa, pancreas, cecal tonsils, small intestine, and caecum) and lengths (small intestine and caecum) were not affected in the bulk iron or IONP-supplemented birds compared to the control birds (Table 3).

### 3.3. Effects of Iron Oxide Nanoparticles Supplementation on the Blood Biochemistry of Broilers

The serum glucose and lipid profile (Cholesterol, Triglycerides, HDL, and LDL) remained unaffected with the supplementation of bulk iron or IONPs. Aspartate aminotransferase levels were higher (*p* < 0.05) in the birds supplemented with 20 and 40 mg/Kg IONPs compared to the control and bulk iron-supplemented groups. Alanine aminotransferase levels remained unchanged with the supplementation of IONPs compared to the control group. Creatine kinase was higher (*p* < 0.05) in the 80 mg/Kg IONP-supplemented group compared to the control group. While serum lactate dehydrogenase remained unaffected with the supplementation of IONPs compared with the control group. Significant quadratic and linear trends were found in creatine kinase and lactate dehydrogenase, respectively. Serum creatinine, urea, and uric acid levels remained unaffected with the supplementation of bulk iron or IONPs. Also, no changes in total proteins, albumin, and globulins were observed (Table 4).

### 3.4. Effects of Iron Oxide Nanoparticles Supplementation on Redox Status of Various Tissues of Broilers

The serum, liver, and muscle MDA levels remained unchanged (*p* > 0.05) with the supplementation of IONPs. Also, no changes (*p* > 0.05) in catalase levels were observed in the studied tissues with IONP supplementation compared with the control group (Table 5).

### 3.5. Effects of Iron Oxide Nanoparticles Supplementation on Meat Quality Attributes in Broilers

The carcass traits (carcass weight, eviscerated weight, dressing percentage, and carcass yield) remained non-significant in all the supplemented groups compared with the control group (Table 6).

The pH_45min_ remained the same in all the supplemented groups compared with the control group. However, the pH_24h_ was decreased (*p* < 0.05) in the birds supplemented with 40 and 60 mg/kg IONPs compared to the bulk and control groups. The cooking loss and WHC remained unaffected with IONP supplementation (Table 7).

The muscle fascicle diameter of the pectoral muscle was significantly increased in the birds supplemented with 20 mg/kg of IONPs compared with the control group. The muscle fiber diameter of the pectoral muscle was higher (*p* < 0.05) in the birds supplemented with 40 mg/kg IONPs compared to the bulk and control groups. In addition, both linear and quadratic trends were observed in muscle fiber diameter. The muscle fiber density was significantly higher in the birds supplemented with 20 mg/kg IONPs compared with the bulk group (Table 7).

No changes in meat color in terms of L* (Lightness), a* (Redness), b* (Yellowness), H (Hue), and C (Chroma) were observed with the bulk iron or IONP supplementation compared to the control group (Table 8).

### 3.6. Effects of Iron Oxide Nanoparticles Supplementation on Jejunal Morphology in Broilers

The villus height of the jejunum was significantly increased with 60 mg/kg IONP supplementation compared to the other supplemented and control groups. The villus width was decreased (*p* < 0.001) in the birds supplemented with 40 mg/kg IONPs compared with the other supplemented and control groups. The crypt depth and villus surface area were significantly higher in the birds supplemented with 60 mg/kg IONPs compared to all the other supplemented and control groups. The villus-height-to-crypt-depth ratio of the jejunum in the birds supplemented with IONPs was lower (*p* < 0.001) compared to the bulk iron and control groups. Moreover, a significant linear effect was observed in villus width and villus surface area whereas both linear and quadratic trends were found in the crypt depth and villus-height-to-crypt-depth ratio (Table 9).

## 4. Discussion

A study on the possibility of iron nanoparticles being used in poultry diets was previously published by Nikonov et al. [1]. Similarly, it has already been established that feed additives in nano form have a variety of impacts, including the capacity to increase immunity and development, act as antioxidants, and, eventually, to provide minimal quantity application in contrast to their bulk counterparts. The chemical form of a mineral source has a substantial impact on mineral absorption and bioavailability. This concept leads to the idea that iron nanoparticles can potentially be an excellent alternative to available microelement-based preparations [24]. Many previous studies have observed increased bodyweight in response to dietary supplementation of nano-Fe in quails and broilers [25,26]. Considering our study, an increase in bodyweight gain was seen in birds fed with 60 mg/kg IONPs. This was in agreement with the findings of Miroshnikov et al. [27], who reported a significant increase in bodyweight of 13.9%, on day 28, compared to the control, with the supplementation of 4 mg/kg of feed nano-Fe. However, the nano-Fe was supplemented with a complex of amino acids that could have led to better weight gain. A similar increase in bodyweight was also reported by Rehman et al. [3], who supplemented broilers with a combination of iron nanoparticles and endoxylanase. An increase in bodyweight gain was also reported in broilers supplemented with nano-Fe reared under heat stress [28]. The trend in improvement of FCR in the present study was observed in the group of birds fed with 20 mg/kg IONPs during the first two weeks. However, at the end of the experiment, better FCR was recorded in the birds fed with 60 mg/kg of IONPS. Our results are in line with the results reported by Rehman et al. [3], who revealed better FCR values in broilers supplemented with iron nanoparticles in combination with endoxylanase. The better FCR in our study might be due to the improvement in jejunal histology by the IONP supplementation, which increased the feed absorption and hence production performance. In the present study, there were no effects on the relative organ weights of the liver, gizzard, spleen, and bursa by the dietary intake of IONPs. Similar findings were also reported in a different investigation using heat-stressed broilers, where adding nano-iron supplementation had no influence on the relative weights of the liver and gizzard [28].

No significant changes in the glucose, cholesterol, triglycerides, LDL, and total protein levels were seen as a result of the supplementation of iron-glycine in broilers [29]. The present study also revealed that IONP supplementation does not alter serum glucose, triglycerides, and protein levels. Our results reported no effects of IONP supplementation on the serum creatinine, urea, and uric acid levels of the broilers. In agreement with our results, Rahdar et al. [30] also reported no discernible differences in the serum urea nitrogen and creatinine levels observed in rats supplemented with 40 mg/kg bodyweight iron oxide nanoparticles compared to the control group. Creatine kinase is mainly stored in the muscle when stress occurs, this enzyme may be quickly released into the blood [31]. In our study, the level of creatine kinase was highest in broilers supplemented with 80 mg/kg of IONPs, suggesting that the higher level of IONPs might have induced stress in the muscle and resulted in an increase in the serum creatine kinase levels. However, the studied meat-quality parameters, in the present study, were not disturbed by this dose of IONPs. Similarly, levels of aspartate aminotransferase increased in the 20 and 40 mg/Kg IONP-supplemented groups. This could be due to the cytotoxic effects of IONPs at low doses [32].

Malondialdehyde (MDA) is a byproduct of polyunsaturated fatty acids peroxidation in cells that acts as an indication of changes in membrane fluidity and fragility [33]. In our study, the serum, liver, and muscle MDA levels were not affected by the IONP supplementation. Our results are in line with the study conducted by Han et al. [34], who reported no changes in the serum and liver MDA levels in broilers supplemented with iron chelated with lysine and glutamine. Iron nanoparticle supplementation decreased the hepatic MDA levels in broilers subjected to heat stress, indicating the antioxidant potential of IONPs, and suggesting the need for IONP supplementation during heat stress conditions [28]. Gou et al. [35] found that dietary iron supplementation in the form of ferrous gluconate enhanced MDA content in the jejunal mucosa of Chinese yellow broilers without affecting plasma MDA levels. Catalase, a common tetrameric heme-containing enzyme is responsible for catalyzing the conversion of two H_2_O_2_ molecules into oxygen and water [36]. In our study, the serum, liver, and muscle catalase activity remained unaffected with the supplementation of IONPs. Han et al. [34] reported increased serum catalase activity in broilers supplemented with 40 and 80 mg/kg iron chelated with lysine and glutamic acid. However, no effects on the liver, heart, and kidney catalase levels were reported in Arbo Acres broilers supplemented with 97–136 mg FeSO_4_/kg of feed [37].

The present study reported no effect on carcass characteristics with the supplementation of IONPs. Kwiecień et al. [29] also reported no effect of iron sulphate or iron chelated with glycine supplementation on the dressing percentage of broilers. Contrary to our results, Behroozlak et al. [38] reported an increased carcass yield with the supplementation of iron hydrogen phosphate nanoparticles in broilers. This might be due to a difference in species and also the size and nature of nanoparticles being used. Meat pH is an important qualitative characteristic of meat that affects the appearance of the meat as well as its juiciness [39]. Normal 15 min postmortem pH levels in chickens range from 6.2 to 6.5, but normal ultimate pH values measured after 24 h range up to 5.8. If the pH value after 15 min is low (below 6.0) while the muscle is still warm, then the proteins are denatured, resulting in a reduced water-holding capacity and meat decolorization [40]. In the current study, a significant drop in the final pH of the pectoral muscle, measured after 24 h was observed in birds supplemented with 40 mg/kg IONPs compared with the 20 mg/kg IONP- and 80 mg/kg bulk iron-supplemented birds; however, the pH values were within the range of pH reported by Duclos et al. [40]. Our results are in line with the study conducted by Berri et al. [41], which compared the experimental and commercial lines of broilers and reported that the ultimate pH, measured 24 h after slaughtering, ranged from 5.5 to 6.4 on average. In our study, the water-holding capacity and cooking loss of breast muscle were not affected by the IONP supplementation. Meat color is one of the major characteristics people consider, especially in boneless items, and it is an indicator of meat quality. Texture and water-holding capacity are two more crucial meat quality attributes that influence consumer preferences [42]. Metabolic products, such as oxygen-based reactive species, antioxidants, lactic acid, or phosphoric acid, can cause myoglobin infiltration into the circulation and increase the L* (lightness) of meat [43]. PSE (Pale, soft, and exudative) meat is one of the hurdles to the poultry meat industry. Studies in the literature regarding the effects of iron oxide nanoparticles supplementation in broilers on meat color are scarce. Our results showed that meat color determined in terms of L*, a*, b*, and H remained the same with either bulk iron or IONP supplementation. Many studies have reported that iron can boost the genesis of red muscular fibers as well as skeletal muscle protein synthesis. Therefore, adequate iron in the muscle may be a viable therapeutic target [44]. Keeping in view this role of iron, we hypothesized that supplementation of IONPs will maintain the muscle iron stores, which is a pre-requisite for muscle protein synthesis. Muscle fiber structure and organization become more defined and assembled during muscle growth, and muscle fiber diameter steadily rises with the growth of broilers [45]. In our study, the pectoral muscle histological attributes varied among the experimental groups with the supplementation of IONPs. The role of IONPs relevant to the muscle histology in broilers has not been reported on yet.

In birds, an improvement in the gut’s mucosal shape is considered a sign of health and growth. The intestinal villi and microvilli significantly improve the absorption of nutrients across the gut [46]. Iron is absorbed in the small intestines as Fe^2+^, which combines with apoferritin to form ferritin. Iron is transported by transferrin to several organs, including the liver, where it is stored as ferritin [32]. No studies in the literature are available on the effect of IONPs on the intestinal histology of broilers. Supplementation of 4 mg/kg nano-iron alone and in combination with methionine induced histopathological lesions in the intestine of heat-stressed broilers [28]. However, the cecal microbiome remains undisturbed with supplementation of 8 mg/kg iron nanoparticles in broiler chickens [47]. In the present study, the birds supplemented with 60 mg/kg IONPs exhibited an increased villus height, crypt depth, and villus surface area of the jejunum. These improvements in the histological attributes might have resulted in better bodyweight gain and an improved feed conversion ratio of the broilers. Intestinal histological parameters, such as villus height, crypt depth, villus surface area, and villus-height-to-crypt-depth ratio represent the overall gut health. Longer villi have a larger luminal surface, which allows for better absorption of the nutrients needed for animal development [48].

## 5. Conclusions

In conclusion, iron oxide nanoparticles supplementation improved the feed-conversion ratio and jejunal morphology. Better meat quality in terms of improved pectoral muscle pH_24h_ and histological attributes was achieved by the IONP supplementation, indicating good quality meat for human consumption. Moreover, viscera characteristics, serum glucose, lipid profile, kidney functions, and antioxidant status were not influenced by IONP supplementation.

## Figures and Tables

**Figure 1 animals-14-00099-f001:**
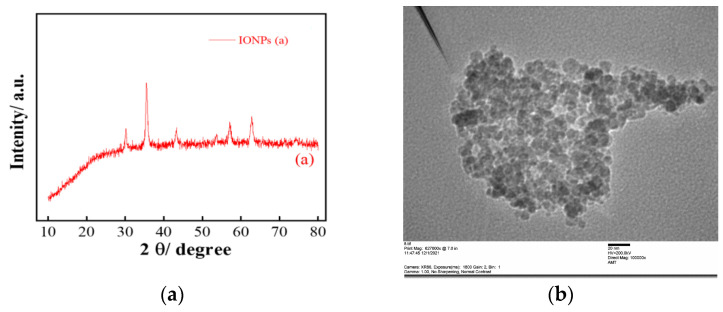
Characterization of iron oxide nanoparticles: (**a**) X-ray diffraction graph shows the presence of the peaks of IONPs; and (**b**) transmission electron microscopy was conducted to confirm the morphology and size of the IONPs.

**Table 1 animals-14-00099-t001:** Composition of the starter and grower diets.

Ingredients (g/kg)	Starter Feed (Day-1 to Day-20)	Grower Feed (Day-21 to Day-35)
Corn	576.8	605.3
Soybean meal	390.8	343.3
Soya oil	0	21.6
Di-Calcium Phosphate	8	5.7
Sodium Chloride	3.7	3.6
Sodium bicarbonate	1	1
Lysine	2.6	2.9
Methionine	3.3	3
L-Threonine	1	1
Choline	1	1
Mineral premix ^1^	1	1
Vitamin premix ^2^	0.5	0.5
Lime	10.3	10.1
Total	1000	1000
	Nutrient contents	
Metabolizable energy (Kcal/Kg)	2722.6	2890.6
Crude protein	23.08	21.16

^1^ Supplied minerals per kg of the feed: Na (NaCl), 3.65 g; K (KCl), 70 g, Mg (MgSO_4_), 6 g; I (KI), 40 mg; Cu (CuSO_4_), 80 mg; Zn (ZnSO_4_), 228.8 mg; Mn (MnSO_4_), 250 mg; Fe (FeSO_4_ or IONPs), Control = 0 mg, Bulk = 222 mg FeSO_4_, 20-IONPs = 28 mg IONPs, 40-IONPs = 57 mg IONPs, 60-IONPs = 85 mg IONPs, 80-IONPs = 114.5 mg IONPs; Co (CoSO_4_) 0.8 mg; Se, 50 mg; Ca, 168.4 mg to 390.4 mg. ^2^ Feed(kg)-contained vitamins(vit) and minerals: vitamin A, 11,000 IU; vitamin B12, 0.0132 mg; vitamin D3, 2200 IU; vitamin E, 22 IU; choline chloride, 440 mg; riboflavin, 8.8 mg; pantothenic acid, 22 mg; ethoxyquin, 250 mg; menadione, 2.2 mg; pyridoxine, 4.4 mg; folic acid, 1.1 mg; biotin, 0.22; thiamine, 4.4 mg.

**Table 2 animals-14-00099-t002:** Effects of bulk iron and iron oxide nanoparticles supplementation on growth performance of broilers.

Parameters ^5^	Week	Control ^1^	Bulk ^2^	IONPs (mg/kg Feed) ^3^				SEM ^4^	*p*-Value	Linear	Quadratic
				20	40	60	80				
BWG (g)	1st	116	112	115	119	105	109	2.13	0.511	0.267	0.589
	2nd	243	235	241	228	214	238	3.73	0.226	0.176	0.333
	3rd	362	366	390	381	363	381	4.92	0.522	0.442	0.383
	4th	587 ^b^	585 ^b^	559 ^b^	599 ^b^	715 ^a^	574 ^b^	14.02	0.024	0.019	0.010
	5th	496	621	602	614	719	581	22.09	0.095	0.087	0.079
FI (g)	1st	137	134	135	141	133	144	2.07	0.670	0.384	0.484
	2nd	358	339	334	361	342	360	4.84	0.486	0.479	0.282
	3rd	544	530	578	545	530	552	6.02	0.187	0.967	0.566
	4th	850	839	859	856	858	840	10.33	0.991	0.975	0.678
	5th	1084	1132	1094	1190	1143	1085	14.78	0.779	0.815	0.349
FCR	1st	1.18 ^b^	1.19 ^b^	1.17 ^b^	1.18 ^b^	1.27 ^ab^	1.32 ^a^	0.017	0.031	0.004	0.077
	2nd	1.47	1.44	1.38	1.59	1.59	1.51	0.029	0.077	0.099	0.055
	3rd	1.51	1.44	1.49	1.43	1.46	1.45	0.034	0.873	0.853	0.783
	4th	1.44	1.43	1.54	1.43	1.20	1.46	0.021	0.088	0.364	0.844
	5th	2.18 ^a^	1.82 ^b^	1.82 ^b^	1.94 ^a^	1.58 ^c^	1.86 ^b^	0.045	0.003	0.005	0.007

^a–c^ Different superscript indicates significant difference between the groups (*p* < 0.05). ^1^ Control = Basal diet without iron supplementation; ^2^ Bulk = Basal diet supplemented with 80 mg/kg FeSO_4_; ^3^ 20 = Basal diet supplemented with 20 mg IONPs per kg of feed; 40 = Basal diet supplemented with 40 mg IONPs per kg of feed; 60 = Basal diet supplemented with 60 mg IONPs per kg of feed; 80 = Basal diet supplemented with 80 mg IONPs per kg of feed. ^4^ SEM = Data are presented as the mean and standard error of means. ^5^ BWG = Bodyweight gain; FI = Feed intake; FCR = Feed conversion ratio.

**Table 3 animals-14-00099-t003:** Effects of bulk iron and iron oxide nanoparticle supplementation on relative viscera weights and lengths.

Parameters	Control ^1^	Bulk ^2^	IONPs (mg/kg Feed) ^3^				SEM ^4^	*p*-Value	Linear	Quadratic
			20	40	60	80				
Relative Visceral Weights (%)										
Liver	2.25	2.23	2.15	2.12	2.35	2.21	0.03	0.409	0.853	0.372
Heart	0.54	0.54	0.51	0.55	0.56	0.51	0.01	0.529	0.972	0.656
Gizzard	2.10	2.07	2.07	2.22	2.19	2.05	0.03	0.757	0.739	0.484
Proventriculus	0.45	0.43	0.43	0.48	0.47	0.45	0.01	0.593	0.324	0.894
Spleen	0.10	0.09	0.09	0.10	0.11	0.09	0.00	0.730	0.973	0.981
Bursa	0.20	0.21	0.24	0.24	0.23	0.19	0.01	0.575	0.930	0.076
Pancreas	0.26	0.29	0.25	0.24	0.26	0.26	0.01	0.212	0.324	0.327
Cecal Tonsils	0.04	0.04	0.03	0.03	0.04	0.03	0.00	0.446	0.165	0.958
SI (EW) ^5^	2.57	2.59	2.39	2.54	2.67	2.69	0.04	0.493	0.285	0.224
Cm (EW) ^6^	0.31	0.32	0.33	0.33	0.37	0.32	0.01	0.678	0.403	0.470
Relative Visceral Lengths (%)										
SI ^7^	8.79	8.36	3.40	8.63	9.09	8.36	0.06	0.716	0.969	0.824
Cm ^8^	0.86	0.86	0.79	0.86	0.86	0.79	0.01	0.580	0.596	0.815

^1^ Control = Basal diet without iron supplementation; ^2^ Bulk = Basal diet supplemented with 80 mg/kg FeSO_4_; ^3^ 20 = Basal diet supplemented with 20 mg IONPs per kg of feed; 40 = Basal diet supplemented with 40 mg IONPs per kg of feed; 60 = Basal diet supplemented with 60 mg IONPs per kg of feed; 80 = Basal diet supplemented with 80 mg IONPs per kg of feed. ^4^ SEM = Data are presented as the mean and standard error of means. ^5^ SI (EW) = Small intestine empty weight; ^6^ Cm (EW) = Caecum empty weight; ^7^ SI = Small intestine; ^8^ Cm = Caecum.

**Table 4 animals-14-00099-t004:** Effects of bulk iron and iron oxide nanoparticles supplementation on serum metabolites.

Parameters ^5^	Control ^1^	Bulk ^2^	IONPs (mg/kg Feed) ^3^				SEM ^4^	*p*-Value	Linear	Quadratic
			20	40	60	80				
Glucose (mg/dL)	149.51	146.15	150.05	160.83	167.49	139.73	3.01	0.084	0.660	0.090
Cholesterol (mg/dL)	104.35	111.94	111.52	110.07	115.27	112.20	1.70	0.605	0.670	0.780
Triglycerides (mg/dL)	98.89	96.16	96.59	108.60	106.87	104.45	3.21	0.815	0.295	0.924
HDL (mg/dL)	52.98	56.51	61.00	54.65	55.33	58.41	0.98	0.213	0.384	0.420
LDL (mg/dL)	31.59	36.20	31.92	33.69	38.56	32.09	1.79	0.851	0.674	0.777
ALT (U/L)	3.21	3.68	4.35	3.78	3.59	5.2	0.33	0.545	0.171	0.722
AST (U/L)	16.80 ^b^	17.63 ^b^	29.17 ^a^	27.70 ^a^	25.76 ^ab^	26.43 ^ab^	1.43	0.032	0.012	0.071
CK (U/L)	573.63 ^b^	808.90 ^ab^	741.34 ^ab^	838.73 ^ab^	730.90 ^ab^	1069.53 ^a^	44.64	0.038	0.457	0.037
LDH (U/L)	313.74	369.18	393.75	373.90	354.37	400.99	14.51	0.585	0.003	0.829
Creatinine (mg/dL)	0.18	0.18	0.15	0.23	0.11	0.25	0.02	0.079	0.004	0.107
Urea (mg/dL)	4.38	3.15	3.86	2.91	1.81	1.81	0.29	0.053	0.972	0.217
Uric Acid (mg/dL)	3.12	3.85	3.38	3.06	3.03	3.78	0.90	0.071	0.869	0.322
Total Protein (g/dL)	4.39	4.28	4.17	4.17	4.41	4.31	0.36	0.667	0.926	0.042
Albumin (g/dL)	2.67	2.60	2.73	2.75	2.65	2.67	0.18	0.878	0.008	0.701
Globulins (g/dL)	1.7	1.58	1.44	1.41	1.75	1.64	0.32	0.162	0.224	0.508

^a,b^ Different superscript indicates significant difference between the groups (*p* < 0.05). ^1^ Control = Basal diet without iron supplementation; ^2^ Bulk = Basal diet supplemented with 80 mg/kg FeSO_4_; ^3^ 20 = Basal diet supplemented with 20 mg IONPs per kg of feed; 40 = Basal diet supplemented with 40 mg IONPs per kg of feed; 60 = Basal diet supplemented with 60 mg IONPs per kg of feed; 80 = Basal diet supplemented with 80 mg IONPs per kg of feed. ^4^ SEM = Data are presented as the mean and standard error of means. ^5^ HDL = High density lipoproteins; LDL = Low density lipoprotein; ALT = Alanine aminotransferase; AST = Aspartate aminotranferase; CK = Creatine kinase; LDH = Lactate dehydrogenase.

**Table 5 animals-14-00099-t005:** Effects of bulk iron and iron oxide nanoparticles supplementation on redox status.

Parameters	Control ^1^	Bulk ^2^	IONPs (mg/kg Feed) ^3^				SEM ^4^	*p*-Value	Linear	Quadratic
			20	40	60	80				
Malondialdehyde (µmol/L or kg)										
Serum	6.33	6.16	6.80	6.24	6.58	6.86	0.29	0.984	0.640	0.917
Liver	7.37	9.70	9.56	6.24	7.80	6.19	0.45	0.076	0.100	0.195
Muscle	6.32	6.85	6.56	6.54	6.36	7.15	0.57	0.083	0.002	0.683
Catalase (KU/L or kg)										
Serum	6.16	7.88	5.21	5.80	7.88	7.76	0.40	0.186	0.289	0.269
Liver	21.17	12.39	22.83	13.97	24.79	18.69	1.59	0.126	0.083	0.577
Muscle	9.65	9.20	8.7	8.65	10.06	9.09	0.43	0.943	0.973	0.639

^1^ Control = Basal diet without iron supplementation; ^2^ Bulk = Basal diet supplemented with 80 mg/kg FeSO_4_; ^3^ 20 = Basal diet supplemented with 20 mg IONPs per kg of feed; 40 = Basal diet supplemented with 40 mg IONPs per kg of feed; 60 = Basal diet supplemented with 60 mg IONPs per kg of feed; 80 = Basal diet supplemented with 80 mg IONPs per kg of feed. ^4^ SEM = Data are presented as the mean and standard error of means.

**Table 6 animals-14-00099-t006:** Effects of bulk iron and iron oxide nanoparticles supplementation on carcass traits.

Parameters	Control ^1^	Bulk ^2^	IONPs (mg/kg Feed) ^3^				SEM ^4^	*p*-Value	Linear	Quadratic
			20	40	60	80				
Carcass Weight (g)	1659	1761	1767	1684	1654	1730	28.85	0.731	0.934	0.659
Eviscerated Weight (g)	1362	1415	1468	1351	1342	1382	20.65	0.505	0.582	0.504
Dressing Percentage	69.88	68.58	71.54	68.93	69.46	68.09	0.46	0.325	0.341	0.330
Carcass Yield (%)	84.73	85.40	85.90	85.72	85.44	85.28	0.29	0.915	0.671	0.295

^1^ Control = Basal diet without iron supplementation; ^2^ Bulk = Basal diet supplemented with 80 mg/kg FeSO_4_; ^3^ 20 = Basal diet supplemented with 20 mg IONPs per kg of feed; 40 = Basal diet supplemented with 40 mg IONPs per kg of feed; 60 = Basal diet supplemented with 60 mg IONPs per kg of feed; 80 = Basal diet supplemented with 80 mg IONPs per kg of feed. ^4^ SEM = Data are presented as the mean and standard error of means.

**Table 7 animals-14-00099-t007:** Effects of bulk iron and iron oxide nanoparticles supplementation on physical and histological traits of pectoral muscle.

Parameters ^5^	Control ^1^	Bulk ^2^	IONPs (mg/kg Feed) ^3^				SEM ^4^	*p*-Value	Linear	Quadratic
			20	40	60	80				
pH_45min_	6.78	6.80	6.78	6.70	6.87	6.89	0.03	0.697	0.348	0.715
pH_24h_	5.74 ^ab^	5.83 ^a^	5.84 ^a^	5.58 ^b^	5.56 ^ab^	5.75 ^ab^	0.02	0.047	0.170	0.706
WHC (%)	15.24	9.13	11.72	10.31	10.96	10.41	0.69	0.168	0.152	0.190
Cooking loss (%)	25.89	20.43	24.27	22.21	20.00	22.33	0.64	0.061	0.095	0.281
MFD (mm)	0.91 ^b^	1.32 ^a^	1.32 ^a^	1.11 ^ab^	0.96 ^b^	1.04 ^b^	0.35	<0.001	0.161	<0.001
MFbD (µm)	56.50 ^c^	67.19 ^bc^	71.92 ^b^	84.24 ^a^	78.62 ^ab^	69.49 ^bc^	2.24	<0.001	<0.001	<0.001
MFbDe	463 ^a^	381 ^b^	455 ^a^	442 ^ab^	417 ^ab^	411 ^ab^	7.67	0.005	0.210	0.922

^a–c^ Different superscript indicates significant difference between the groups (*p* < 0.05). ^1^ Control = Basal diet without iron supplementation; ^2^ Bulk = Basal diet supplemented with 80 mg/kg FeSO_4_; ^3^ 20 = Basal diet supplemented with 20 mg IONPs per kg of feed; 40 = Basal diet supplemented with 40 mg IONPs per kg of feed; 60 = Basal diet supplemented with 60 mg IONPs per kg of feed; 80 = Basal diet supplemented with 80 mg IONPs per kg of feed. ^4^ SEM = Data are presented as the mean and standard error of means. ^5^ WHC = Water holding capacity; MFD = Muscle fascicle diameter; MFbD = Muscle fiber diameter; MFbDe = Muscle fiber density.

**Table 8 animals-14-00099-t008:** Effects of bulk iron and iron oxide nanoparticles supplementation on meat color of pectoral muscle.

Parameters	Control ^1^	Bulk ^2^	IONPs (mg/kg Feed) ^3^				SEM ^4^	*p*-Value	Linear	Quadratic
			20	40	60	80				
L* (Lightness)	51.21	52.04	55.23	54.04	53.67	53.48	0.41	0.052	0.060	0.028
a* (Redness)	13.87	13.68	12.79	12.62	12.79	13.73	0.24	0.472	0.482	0.075
b*(Yellowness)	11.08	10.61	12.63	12.60	12.37	11.95	0.26	0.107	0.065	0.122
H (Hue)	38.64	37.8	44.57	45.11	43.90	41.23	0.94	0.088	0.089	0.046
C (Chroma)	17.94	17.40	18.30	17.90	17.83	18.31	0.20	0.882	0.482	0.624

^1^ Control = Basal diet without iron supplementation; ^2^ Bulk = Basal diet supplemented with 80 mg/kg FeSO_4_; ^3^ 20 = Basal diet supplemented with 20 mg IONPs per kg of feed; 40 = Basal diet supplemented with 40 mg IONPs per kg of feed; 60 = Basal diet supplemented with 60 mg IONPs per kg of feed; 80 = Basal diet supplemented with 80 mg IONPs per kg of feed. ^4^ SEM = Data are presented as the mean and standard error of means.

**Table 9 animals-14-00099-t009:** Effects of bulk iron and iron oxide nanoparticles supplementation on jejunal morphology.

Parameters ^5^	Control ^1^	Bulk ^2^	IONPs (mg/kg Feed) ^3^				SEM ^4^	*p*-Value	Linear	Quadratic
			20	40	60	80				
VH (µm)	912.54 ^b^	877.72 ^b^	756.11 ^c^	672.63 ^c^	1189.83 ^a^	934.45 ^b^	17.17	<0.001	0.752	0.691
VW (µm)	90.35 ^ab^	91.77 ^a^	81.21 ^b^	69.15 ^c^	87.21 ^ab^	80.74 ^b^	1.14	<0.001	<0.001	0.341
CD (µm)	121.62 ^c^	119.71 ^c^	156.84 ^b^	111.53 ^c^	185.82 ^a^	152.25 ^b^	2.65	<0.001	<0.001	<0.001
VSA (mm^2^)	0.26 ^b^	0.25 ^b^	0.19 ^c^	0.15 ^d^	0.32 ^a^	0.24 ^b^	0.006	<0.001	<0.001	0.770
VH:CD	7.78 ^a^	7.41 ^a^	4.82 ^c^	6.09 ^b^	6.46 ^b^	6.20 ^b^	0.12	<0.001	<0.001	<0.001

^a–c^ Different superscript indicates significant difference between the groups (*p* < 0.05). ^1^ Control = Basal diet without iron supplementation; ^2^ Bulk = Basal diet supplemented with 80 mg/kg FeSO_4_; ^3^ 20 = Basal diet supplemented with 20 mg IONPs per kg of feed; 40 = Basal diet supplemented with 40 mg IONPs per kg of feed; 60 = Basal diet supplemented with 60 mg IONPs per kg of feed; 80 = Basal diet supplemented with 80 mg IONPs per kg of feed. ^4^ SEM = Data are presented as the mean and standard error of means. ^5^ VH = Villus height; VW = Villus width; CD = Crypt depth; VSA = Villus surface area; VH:CD = Villus-height-to-crypt-depth ratio.

## Data Availability

The data presented in this study are available on request from the corresponding author. The data are not publicly available due to privacy policy of the institute.

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
