# Peer review of "Effects of Iron Oxide Nanoparticle Supplementation on the Growth Performance, Serum Metabolites, Meat Quality, and Jejunal Basal Morphology in Broilers"

_animals, 2023, doi:10.3390/ani14010099_

Round 1

Reviewer 1 Report

Comments and Suggestions for Authors

I have read carefully the manuscript entitled “Effects of iron oxide nanoparticles supplementation on the growth performance, serum metabolites, meat quality, and jejunal micro-architecture in broilers” submitted to Animals as an original article. Some points should be addressed before the manuscript can be considered for publication.

The main concerns are as follow:

  1. The introduction needs to include a (short) description of the study's novelty.
  2. Was there an overnight fasting period for the birds before slaughter? This detail is crucial as it can significantly impact various measurements, including body weight, liver redox status, viscera measurements, jejunal histology, and blood serum biochemistry. Fasting can have both positive and negative effects on these measurements, depending on whether they should be conducted on fasted birds or not.
  3. Redox status indices should be expressed per gram of protein, not per kilogram of the sample.
  4. Since the experimental factor was introduced through the diet, the basic experimental unit should be the replicate pen. As two birds per replicate were analyzed, all traits should be averaged per replicate before conducting ANOVA (a simpler method) or the replicate pen should be included in the statistical model as a mixed variable (a more complex, but appropriate method).
  5. Adding representative images of muscle/jejunum in the results section could be beneficial, especially since significant differences in calculated traits were observed between groups.

Specific comments:

1.     Title Suggestion: I recommend changing "micro-architecture" to "basal morphology" for more precise terminology. This study assesses basic morphological characteristics of intestine sections, including villus height, width, crypt depth, and the villus-to-crypt ratio (VC:DC), which are fundamental parameters. There are numerous characteristics and indices that could be assessed for intestine sections (for muscosa, submuscosa, muscularis, and serosa).

2.     L47: The term "poultry" is too broad. Could you specify which poultry species were studied?

3.     L48-52: This section is unclear and could benefit from rephrasing for better readability and coherence.

4.     L336-337: Either remove this reference or replace it with a more relevant one.

5.     L88: Please clarify the method used for calculating the particle size of IONPs.

6.     Fig 1: Briefly describe what the XRD diffractogram reveals about the IONPs. At least provide a short explanation in the figure caption.

7.     L92: Which specific breed of chickens was used in this study?

8.     L97: Why was the bulk group supplemented with 80 mg of Fe? How does this dosage align with nutritional recommendations for these broilers? Additionally, how does this relate to the 106 ppm dose mentioned in L47 as optimal for poultry?

9.     Table 1: Could you explain what the abbreviation DCP stands for?

10.  L113: List the traits that were measured in the study.

11.  L120: Please specify which section of the jejunum was analyzed.

12.  L137: AMDCC  - Is this an Animal Models of Diabetic Complications Consortium mentioned? Please provide a reference for the protocol describing MDA determination.

13.  L162 and 167: Describe the histological measurements of muscle/jejunum in greater detail, including the method of measurement, the number of measurements per sample, and how these measurements were incorporated into the statistical model. Additionally, explain how the villus surface area was calculated.

14.  L175: Define the experimental unit. Confirm whether ANOVA assumptions were checked prior to analysis. Also, clarify if the control group (0 IONPs) was included in the polynomial contrast analysis.

15.  Table 2: Significant linear and quadratic trends observed at week 4 for BWG and at week 1 for FCR are not discussed in the text. Similar trends observed for blood serum biochemistry (Table 4) and other tables should be addressed. If polynomial trends are not mentioned in results section nor discussed, their inclusion needs justification.

16.  Discuss potential cytotoxicity of iron nanoparticles and their influence on gut microbiome. Numerous studies on this topic should be referenced (e.g., DOI: 10.1007/s11356-022-19156-4, 10.1007/s11250-022-03130-w, 10.1007/s11356-018-1991-5).

Reviewer 2 Report

Comments and Suggestions for Authors

This manuscript studies the use of iron oxide nanoparticles as feed additive in poultry. The effects were investigated on growth performance, organ growth, serum biochemical parameters, redox status, gut morphology, and meat quality in broilers.  

I can say this paper is suitable for publication with minor changes. I would like to recommend the following suggestions to you.

General comments 

The manuscript is well-structured and clearly written. The research design is appropriate with the proposed objectives, the results are clearly described and discussed. 

Specific comments are discussed below:

  • Line 37. Define abbreviations in the text, e.g. FCR.

  • Figure 1b. If possible, improve resolution.

  • There are @ throughout the text. Please check if it is correct.

  • Line 96. Why did you choose FeSO4 80 mg/kg as control? What are the differences between iron supplements (sulfate or nanoparticles)?

  • Have you analyzed the toxicity of the nanoparticles? 

  • Line 112. Please define feed intake, body weight gain and feed conversion ratio.

  • Line 116. Why did you sample (only) on the 35th day?

  • Line 139. Catalase levels or activity?

  • Discussion. Do you think it would be easy to scale up the production of nanoparticles and their application? What is the real cost? It would be interesting to incorporate it in the manuscript.

  • Line 316-320. There is too much difference between the doses of each study to compare the results (15 fold: 60 vs 4 mg/kg)

  • Line 331. typing mistake (absoprtion)

  • Line 336. Please revise the sentence

  • Line 362. The sentence about catalase seems lost in the text. Please rephrase or delete it.

  • Line 364. It should say activity instead of catalase concentration

  • Line 404. What is your hypothesis about the role of IONPs in muscle histology of broilers?

  • Line 441. Check the references for correct formatting. Please also check all references are present in both the text and the list.

Comments on the Quality of English Language

A minor spell and style checker is required.

Reviewer 3 Report

Comments and Suggestions for Authors

The proposal presented here contributes to the knowledge and importance of the use of iron in poultry diets; however, I suggest discussing in more depth the following questions and not only indicate that our results are similar. 

Why in the study the treatments supplemented with 60 mg/kg of IONPs had an increase in weight, how this particle influences metabolism and mass conversion regarding their other treatments.

To deepen, if it is good or bad that there were no changes in the biochemical studies in the different treatments. 

Why the Nanoparticles generated damage in tissue to what is attributed that the creatine synthase is high, chemically what is happening. 

In general, you need to review the discussion section to support your results and not just make comparisons with other studies. 

Verify its conclusion, showing the importance of including iron oxide nanoparticles in the poultry diet, as well as indicating if they can have an effect on humans when consuming this meat.

Comments on the Quality of English Language

Review of some sentences and style correction 

Round 2

Reviewer 1 Report

Comments and Suggestions for Authors

Comments 3. Redox status indices should be expressed per gram of protein, not per kilogram of the sample.

Response 3. Thank you for highlighting this point. The error has been rectified by correcting the units in Table 5.

Revised Comment 3: I do not understand the corrections made. The authors merely changed the unit from umol/l or kg of tissues to umol/ml or g, claiming it is now expressed per g of protein and not per kg of tissue. This is obviously incorrect. Ignoring the absurdity of such a change (which would imply a thousandfold increase in the concentration of the determined indices), it is still unacceptable because it means that the authors simply swapped units instead of recalculating the concentration of the indices based on the determined protein concentration in the analyzed samples. This is unacceptable.

Comments 19. L175: Define the experimental unit. Confirm whether ANOVA assumptions were checked prior to analysis. Also, clarify if the control group (0 IONPs) was included in the polynomial contrast analysis.

Response 19: Thank you for highlighting this point. The asked queries are addressed and added to the manuscript in line 209-212.

Revised Comment 19: The revised manuscript still lacks information on whether the control group (0 IONPs) was included in the polynomial contrast analysis along with the 20-80 IONPs groups.

Reviewer 3 Report

Comments and Suggestions for Authors esteemed authors I appreciate the response to my comments and their attention to their manuscript.
